# A Comprehensive Review of Artificial Intelligence-Based Algorithms for Predicting the Remaining Useful Life of Equipment

**DOI:** 10.3390/s25144481

**Published:** 2025-07-18

**Authors:** Weihao Li, Jianhua Chen, Sijuan Chen, Peilin Li, Bing Zhang, Ming Wang, Ming Yang, Jipu Wang, Dejian Zhou, Junsen Yun

**Affiliations:** 1Shenzhen Key Laboratory of Nuclear and Radiation Safety, Institute for Advanced Study in Nuclear Energy & Safety, College of Physics and Optoelectronic Engineering, Shenzhen University, Shenzhen 518060, China; l219153833@163.com (W.L.); 2310456012@email.szu.edu.cn (J.C.); 2410236002@mails.szu.edu.cn (P.L.); mingyang@szu.edu.cn (M.Y.); wangjipu@outlook.com (J.W.); q2440772442@163.com (D.Z.); seven1534517441@163.com (J.Y.); 2State Key Laboratory of Nuclear Power Safety Technology and Equipment, China Nuclear Power Engineering Co., Ltd., Shenzhen 518172, China; xiaohan1123@163.com (B.Z.); wangming25436@163.com (M.W.)

**Keywords:** artificial intelligence (AI), remaining useful life (RUL), data-driven analysis

## Abstract

In the contemporary big data era, data-driven prognostic and health management (PHM) methodologies have emerged as indispensable tools for ensuring the secure and reliable operation of complex equipment systems. Central to these methodologies is the accurate prediction of remaining useful life (RUL), which serves as a pivotal cornerstone for effective maintenance and operational decision-making. While significant advancements in computer hardware and artificial intelligence (AI) algorithms have catalyzed substantial progress in AI-based RUL prediction, extant research frequently exhibits a narrow focus on specific algorithms, neglecting a comprehensive and comparative analysis of AI techniques across diverse equipment types and operational scenarios. This study endeavors to bridge this gap through the following contributions: (1) A rigorous analysis and systematic categorization of application scenarios for equipment RUL prediction, elucidating their distinct characteristics and requirements. (2) A comprehensive summary and comparative evaluation of several AI algorithms deemed suitable for RUL prediction, delineating their respective strengths and limitations. (3) An in-depth comparative analysis of the applicability of AI algorithms across varying application contexts, informed by a nuanced understanding of different application scenarios and AI algorithm research. (4) An insightful discussion on the current challenges confronting AI-based RUL prediction technology, coupled with a forward-looking examination of its future prospects. By furnishing a meticulous and holistic understanding of the traits of various AI algorithms and their contextual applicability, this study aspires to facilitate the attainment of optimal application outcomes in the realm of equipment RUL prediction.

## 1. Introduction

Against the background of China’s “Made in China 2025,” the United States’ “Advanced Manufacturing Partnership Program,” Germany’s “Industry 4.0 Strategy Plan,” the United Kingdom’s “UK Industry 2050 Strategy,” and Japan’s “Super Smart Society 5.0 Strategy,” and powered by technologies such as the Internet of Things, big data, and cloud computing, manufacturing enterprises are gradually transforming their production workshops toward automation, digitalization, and intelligence. Therefore, equipment maintenance management work has become an urgent concern for manufacturing enterprises. To ensure the safe and reliable operation of equipment systems in the current big data era, it is of great significance to study data-driven fault prediction and health management (PHM) methods for equipment systems under complex working condition application scenarios. The basic structure of PHM is shown in Figure 1, which includes the prediction of the remaining useful life (RUL) of equipment, which is very important.

The RUL of equipment can be defined as “the time length from the current moment to the end of equipment service life,” expressed as $l_{t_{cur}}=t_{end}-t_{cur}$, where $t_{end}$ represents the moment when the equipment fails, $t_{cur}$ represents the current moment, and $l_{t_{cur}}$ represents the RUL of the equipment at the $t_{cur}$ moment. In some of the literature, the RUL of equipment is also defined as “the time length from the equipment health state to reach the failure threshold (FT),” expressed as $l_{t_{cur}}=inf(l:x\left(t_{cur}+l\right)>\gamma )$, where inf(·) represents the inferior limit of a variable, $x\left(t_{cur}+l\right)$ is the health state at $t_{cur}+l$ with l≥0, and γ is the FT. It is important to note that since FT is affected by multiple variable resources, it should be represented using a probability distribution rather than a constant line [1].

The core idea of equipment RUL prediction is to determine the effective remaining usage time or the probability distribution and mathematical expectation of remaining usage time by establishing the degradation mapping relationship of equipment based on an equipment degradation mechanism model, or the use of data monitoring and artificial intelligence methods, and comparing this with the failure threshold. Based on this idea, equipment RUL prediction methods can be divided into three categories: (a) physical model methods, (b) data-driven methods, and (c) model and data fusion methods. The fusion method generally refers to the process of combining various signals, data, or model results using certain techniques or algorithms to obtain a more accurate, comprehensive, or optimized outcome. Due to the complex structure, diverse failure models, and uncertain operating conditions of some systems, it is difficult to establish physical failure models. Data-driven methods that do not require prior knowledge or a complex physical modeling process have become a research hotspot in recent years. In recent years, with the continuous progress of computer hardware and machine learning algorithms, artificial intelligence research has made great progress. Especially with the rapid development of the Internet of Things, a large number of different types of sensors can easily collect a large amount of historical and real-time data from machine equipment; this allows for the use of artificial intelligence techniques to learn from big data and automatically diagnose the health status of machine equipment and predict its future health development trend. In this situation, diagnosis systems based on traditional technologies are being replaced by those based on artificial intelligence, thereby improving the efficiency of equipment health management.

Equipment RUL prediction based on AI methods generally consists of three main technical processes: data acquisition, health factor HI construction, and RUL prediction. In recent years, many studies have been carried out based on each of these technical processes, but at present, most focus on the latter process (i.e., RUL prediction) for in-depth research and discussion, lacking a review that effectively covers each process for equipment remaining life prediction based on AI methods. Therefore, this study reviews equipment RUL based on AI methods according to these technical processes.

This RUL prediction method framework first undergoes the “sensor data→HI construction” process, where sensor data are collected, preprocessed, and then features are extracted and screened to construct a health indicator (HI). It then enters the “HI construction→RUL prediction” phase, where an AI method is selected. After model training, optimization, evaluation, and validation, RUL prediction is performed. Based on the prediction accuracy, if it does not meet the standard, the model is optimized; if it meets the standard, it provides support for equipment maintenance decisions, realizing a complete prediction process from data collection to guiding maintenance and facilitating the predictive maintenance of equipment. A framework diagram of RUL is shown in Figure 2.

The rest of this paper is organized as follows. Section 2 discusses the classification of data and the main data sources used for RUL prediction research. Section 3 introduces the main methods for constructing the health factor HI. Section 4 discusses AI methods and evaluation indicators applied to RUL prediction. Section 5 provides a case study based on the applicability of the algorithms. Finally, Section 6 gives conclusions and discusses the challenges and prospects of RUL prediction technology based on AI methods at the present stage.

## 2. Data Acquisition

The most common method used for equipment data acquisition is to collect real-time data during operation through various sensors (such as temperature sensors, vibration sensors, and pressure sensors). With the continuous progress of sensor technology, modern sensors have higher accuracy, faster sampling rates, and wider application ranges. Modern equipment usually uses multi-modal sensor fusion, equipped with various types of sensors, such as temperature, pressure, vibration, sound, and other sensors. The fusion of these multi-modal sensors can provide multi-dimensional and multi-angle data, which helps to comprehensively understand the state of equipment and make more accurate remaining life predictions. In addition, the large-scale data generated by sensors can be stored, processed, and analyzed by cloud computing technology. This cloud-based data processing mode can realize cross-regional, real-time, and scalable data analysis. In addition to using sensors to obtain real-time data, equipment usually generates log files or saves historical operation data, so data can be obtained through equipment logs and historical records. These data can include equipment operation status, maintenance records, and fault information. Records of regular maintenance and the replacement of parts can be used to judge the degree of aging and usage status of equipment, which is of great significance for predicting its remaining life.

Although the current equipment data acquisition methods are diversified, some complex equipment may transition from health to failure through long-term degradation processes, which may be as short as a few months or as long as several years, leading to the collection of fault data being time-consuming and costly. Moreover, in practice, some equipment is not allowed to run to failure, such as nuclear power units, aircraft engines, and automobile gearboxes, because accidental failure can cause the whole system to shut down or even lead to catastrophic accidents. In addition, due to the involvement of some equipment in military secrets or business competition, few military or commercial institutions that can collect run-to-failure data are willing to disclose their data. Due to these obstacles, the data obtained from different equipment are not all large-sample data. Most of the equipment data in the literature on RUL prediction are not obtained from real industrial equipment. Therefore, the first part of this section classifies the data and discusses the commonly used RUL estimation models for each classification; the second part introduces the equipment and its data used in the literature on RUL prediction, hoping to provide some reference for researchers on the type and acquisition of data.

### 2.1. Data Classification

As shown in Table 1, based on the differences in the information content of the data, this part divides the data into three categories: fully degraded data, partially degraded data, and fault time data. Next, these three types of data are introduced, and three common RUL model estimation methods, along with their related literature, are discussed. The correspondence between the model and the data is shown in Figure 3.

#### 2.1.1. Fully Degraded Data

Fully degraded data refers to data that have undergone the entire degradation process, from normal to failure, in the acquired device data. Since the data state of the device or component being monitored mostly changes with time, the data are of the time series type, which can be predicted using similarity models based on historical data. These models can be divided into local similarity prediction and global similarity prediction. The RUL prediction method is based on local similarity searches for similar historical data with the degradation behavior of the device to be tested, and it follows the notion that local data can reflect the degradation situation. Most researchers believe that the recent state largely reflects the future degradation behavior of the system, so they use the latest data obtained by the system to perform a similarity search. Zio and Maio [2] suggested calculating the fuzzy distance between the latest test trajectory pattern and each historical trajectory model to find similar patterns. Huang et al. [3] proposed an improved prediction method based on trajectory similarity to deal with the uncertainty of prediction. Since each event experienced by the system may transmit degradation information, Lyu et al. [4] proposed an RUL prediction method based on multi-local similarity, which not only considers the information of events that occurred before but also finds high-quality references for RUL prediction. RUL prediction based on global similarity regards all historical data of the device to be tested as a whole and matches similar degradation behaviors for the prediction. Soons et al. [5] proposed a global similarity model by combining Bayesian updates with prior estimation based on similarity and proved that this method is far superior to other techniques in long-term prediction. Liu et al. [6] used principal component analysis (PCA) to construct the HI for similarity matching between test and reference cutting tools, and finally applied the weighted average value of RUL of all similar reference cutting tools to predict the current test cutting tool’s RUL. Yu et al. [7] used a curve matching technique based on similarity to compare the test HI curve obtained from the sensor readings of online instances with the degradation mode constructed in the offline stage to estimate test unit RUL. Some researchers have combined these two models. In the case of an imbalance between accuracy and efficiency in similarity measurement, and given that existing research does not consider global similarity between samples, Gu and Ge [8] proposed a prediction method based on composite similarity and proved its effectiveness and superiority. Liu et al. [9] proposed a new Bayesian-extreme learning machine parameter update algorithm based on Bayesian theory that combines local and similarity methods to eliminate the influence of multiple uncertain sources and accurately predict RUL. Some researchers have also combined similarity models with artificial intelligence methods to improve prediction accuracy. Hou et al. [10] combined supervised learning with similarity-based prediction to improve RUL prediction accuracy. Chen et al. [11] proposed a framework for estimating aircraft engine RUL using full life cycle data and non-fault performance deterioration parameter data based on similarity theory and a support vector machine (SVM). The life prediction method based on similarity theory can predict without establishing a degradation model, which can effectively avoid the difficulties of establishing a degradation model, obtaining degradation model parameters, and model generalization. Even if the amount of data held is limited, it still has good estimation accuracy.

#### 2.1.2. Partially Degraded Data

Partially degraded data refers to a situation where, in some cases, there is no faulty device of the same type, and only part of the data—from the normal state to the fault time and the safety threshold at the fault critical point—is obtained from the acquired data; that is, the known safety threshold that should not be exceeded. In this case, some models can be used to approximate the degradation track according to the device’s degradation data, and then predict the remaining life of the device, which is called the degradation model method. As the degradation model can infer future conditions from historical behavior, it does not need historical data from similar devices and is suitable for predicting objects with degradation data. In [17], Zhang et al. used a degradation model with a random failure threshold to predict the RUL of cutting tools for tool wear processes. For the remaining service life of lithium batteries, Wang et al. [14] proposed a prediction method based on a non-Markovian Brownian motion degradation model. In recent years, some researchers [12,15,16,17] have combined degradation models with artificial intelligence algorithms to improve RUL prediction accuracy and achieve complementary advantages. Compared to the similarity model method, this method collects simpler data but also has limitations. Before using this method, the failure threshold of the device needs to be known. However, the parameters used to characterize degradation are often abstract and difficult to interpret in terms of physical meaning. At the same time, historical failure data are often unavailable, making it difficult to determine the failure threshold. On the other hand, as the prediction length increases, the prediction error of the model also accumulates continuously, resulting in poor long-term prediction accuracy.

#### 2.1.3. Fault Time Data

Fault time data refers to cases where the device does not have historical data leading up to the fault time, and only fault data and some covariates related to RUL are available. In this case, survival analysis is a useful method, also known as the survival model method. This is a statistical method for modeling survival curve (time-to-event) data, which is divided into two specific methods: the reliability survival model and the covariate survival model. The reliability survival model estimates the probability distribution of failure time using the life distribution; that is, it does not need device degradation data—only the life distribution. In contrast, the covariate survival model is a proportional risk survival model that uses life data and covariates to calculate the device’s survival probability. Common proportional risk functions include the exponential regression model, the Weibull regression model, and the Cox regression model, among which the Cox regression model belongs to the proportional risk semi-parametric model. In [18], Ding et al. proposed a new method for reliability evaluation of device state vibration characteristics based on a proportional failure rate model and illustrated the rationality and effectiveness of this method by applying it to bearings. Wang et al. [19] used a method based on kernel principal component analysis (KPCA) and the Weibull proportional hazard model (WPHM) to evaluate the reliability of rolling bearings. Currently, there are also researchers who combine survival models with other methods for RUL prediction. Xu et al. [20] used a reliability covariate model based on a time series chain graph to realize RUL prediction of a turbofan engine. Chu et al. [21] proposed a deep learning method combined with a deep survival model based on the discrete Weibull distribution to estimate RUL.

### 2.2. Equipment Data

At present, PHM technology has become more mature, greatly improving the overall safety and stability of the system, and has been widely used in various fields. The RUL prediction of equipment is an indispensable and important link in RHM, which can provide accurate inspection and maintenance times by predicting the remaining life of equipment and ensuring the safe and reliable operation of equipment. Based on common equipment RUL nowadays, this part introduces three types of equipment and their characteristics: namely, rolling bearings [12,22,23,24,25,26,27,28,29,30,31,32,33,34,35], lithium batteries [27,28,29,30,31], and turbofan engines [32,33,34,35,36], as shown in Table 2. Readers can choose the appropriate equipment as the research object according to their research needs. The three kinds of equipment mentioned above are not only widely used in various fields, such as aerospace, industrial manufacturing, new energy products, and electronic devices, but are also involved in various machines and equipment that may cause different degrees of property loss, safety accidents, or even threaten human life if they fail. Whether from the perspective of equipment cost or safety issues, RUL analysis of the above equipment is essential. Therefore, if we can monitor and evaluate the health status of the equipment in the early stage of degradation and ensure that it is maintained at the best time, we can effectively reduce the probability of accidents, greatly improve the safety and stability of the equipment, avoid unnecessary property loss and maintenance costs, and prevent casualties and disasters.

#### 2.2.1. Rolling Bearings

Rolling bearings are an important part of modern industrial machinery and equipment, which can reduce friction loss. The basic structure of the most common rolling bearing consists of four parts: outer ring, rolling element, cage, and inner ring, as shown in Figure 4. Some bearings also include a grease seal, stop ring, increased number of rolling elements, or reduced cage. The basic structure of rolling bearings is relatively fixed, but it can also be flexible and changeable when used. They are widely used in various industrial equipment. Therefore, by studying the vibration frequency of different parts of bearings, we can effectively determine the health status of equipment and achieve accurate fault prediction. Because a rotating shaft works at a specific speed and load state, it inevitably produces vibration. Many factors affect vibration, which can be roughly divided into internal and external factors, as shown in Figure 5. The external factors are the following:The resonance effect, which occurs when the mechanical equipment operates;The interaction force between parts, caused by the assembly structure of mechanical equipment parts;The slight influence of the working environment on its basic structure, such as humidity, pressure, and temperature, which are also related to internal factors, such as the temperature change caused by friction between rolling elements and inner and outer rings.

The internal factors are the key to affecting vibration, which are the following:1.Forced vibration caused by a machining assembly error or mistake.

In the process of machining rolling bearings, scratches, friction, and uneven smoothness inevitably occur. There is also position deviation in the assembly process, and rolling elements cause complex and random vibration when rotating and sliding.

2.Inherent vibration caused by the bearing structure.

Vibration is caused by the material and structure of the rolling bearing when it rolls. It is also related to the change in gravity and centripetal force on different parts of the inner and outer rings caused by the inclination angle of the bearing. When the bearing is in a normal operational state, it produces regular and definite periodic vibration.

3.Impact vibration caused by mechanical failure.

Bearings often suffer from lubrication failure, overload, impact/collision from the outside world, adhesion, and even fracturing during operation. At the moment of being affected, there is usually a small pulse vibration.

As shown in Figure 6, the operation cycle of rolling bearings can be divided into the normal stage, degradation stage, and failure stage. In the normal stage, rolling bearings are just starting to be used. Their performance indicators are good and can meet normal work requirements. At this stage, bearings are in a healthy state with a low degree of wear, which may be difficult to detect. With increasing running time of the equipment, rolling bearings suffer slight wear, and performance indicators gradually decline, but they can still work normally. At this time, the degree of wear is obvious and can be detected, and the rolling bearing enters the degradation stage. In the degradation stage, the performance indicators of bearings gradually decline with time. According to the specific situation, consideration can be given as to whether to replace the current bearing with a new bearing. With increasing use time, the degree of damage to bearings gradually increases, finally reaching the failure point and entering the failure stage. In the failure stage, the use of the rolling bearing must stop, and it needs to be replaced or repaired. If bearings continue being used in this stage, there will be unpredictable risks.

Therefore, through the above analysis, it can be seen that when bearings are in a normal and good working state, their vibration signal consists of low-level random fluctuations, making it difficult to extract degradation-related information. However, during the degradation stage, with increasing time, the vibration amplitude also increases, sudden pulse signals appear more frequently, and the vibration frequency becomes rich in degradation-related information. Therefore, the RUL prediction of bearings is mainly based on data from the degradation stage to complete failure.

At the same time, because bearings usually work alongside many other parts of a piece of equipment, there is a lot of noise in the information collected by sensors. Denoising of the original signal plays a key role in feature extraction and subsequent prediction, which is key to helping improve the accuracy of the RUL prediction of bearings. Common vibration signal analysis methods include the time domain analysis method, the frequency domain analysis method, and the time–frequency domain analysis method. For the RUL prediction of bearings, besides the vibration analysis method, there are also the temperature analysis method and the oil analysis method. Generally, the vibration analysis method is more common and conducive to information analysis and calculation.

#### 2.2.2. Lithium Batteries

With the increasing fermentation of environmental pollution and the energy crisis, countries around the world have begun to advocate and develop new energy equipment. New energy vehicles have emerged and have been popularized. Power batteries are the power source of such electric vehicles. Among the many types of batteries, compared to lead–acid, nickel–cadmium, and nickel–hydrogen batteries, lithium batteries have the advantages of a long cycle life, a high rated voltage, a high energy utilization rate, and no memory effect. At the same time, they also have the advantages of a lighter battery, a higher specific energy, and a lower self-discharge rate. Lithium batteries are widely used in smartphones, new energy vehicles, and even aerospace fields. Because of the wide application of lithium batteries, their RUL prediction is very important. If they are not replaced or repaired in time, they can cause dangerous accidents such as the spontaneous combustion of a battery and equipment explosions. The main factors affecting the degradation of lithium batteries are their own working mode and the external environment, and the degradation process has high uncertainty.

The internal structure of lithium batteries is shown in Figure 7. When the battery is charged, lithium ions Li+ are released from the positive electrode, and the positive electrode undergoes an oxidation reaction, resulting in an increase in the potential of the positive electrode. Li+ is also reduced to Li and embedded in the negative electrode. When the battery is discharged, Li+ is released from the negative electrode, and the negative electrode undergoes an oxidation reaction. Li+ is also reduced to Li and embedded in the positive electrode. During the charging and discharging process, Li+ moves inside, while electrons move outside the circuit, forming the current. The electrodes and overall cell reactions of common lithium batteries are as follows:

Positive electrode:(1)$$LiMO_{2}⇌{Li}_{1-x}MO_{2}+x{Li}^{+}+xe^{-}$$

Negative electrode:(2)$$C+y{Li}^{+}+ye^{-}⇌{Li}_{y}C$$

Overall cell reaction:(3)$$LiMO_{2}+x/yC⇋x∕y{Li}_{y}C+{Li}_{1-x}MO_{2}$$

In the formula, M represents a metal element, and x and y are the molar capacities of Li based on the electrode material.

Theoretically, according to the chemical reaction formula of lithium batteries, the total number of lithium ions ideally remains unchanged, and the battery does not age during the charging and discharging process of a lithium battery. However, in reality, some other irreversible electrochemical reactions occur in the battery during the charging and discharging process, which cause changes in battery structure and material and make the battery age. In addition, the working environment and usage of lithium batteries also affect the aging of the battery, thus affecting its performance and resulting in the degradation and scrapping of lithium batteries.

There are three main internal factors affecting the aging of lithium batteries:1.Temperature

Although lithium batteries have a better working temperature range than most batteries, their stable working range is only 0~40 degrees Celsius. When the temperature is too high, although the charging and discharging reaction rates increase, the intensified side reactions also consume a lot of the lithium ions, resulting in a decline in battery capacity. Meanwhile, when the temperature is too low, the activity of the materials is greatly reduced, resulting in a reduced reaction rate, an incomplete reaction, and other situations. This also causes lithium ion crystallization attached to the negative electrode material, resulting in increased internal resistance.

2.External stress

External impact and extrusion can not only cause damage and rupture the battery shell, but also lead to short circuits or short-circuit failure. A more serious impact can directly damage battery materials and electrolyte leakage, resulting in the battery being scrapped.

3.Unreasonable charging and discharging rates

Overcharging causes a lot of heat generation in the battery, raises the temperature in a short time, consumes a lot of lithium ions, and may also cause thermal runaway. Meanwhile, over-discharging causes structural damage and damage to battery materials, affecting normal battery reaction.

In addition, lithium-ion batteries show capacity decline over time. One reason is that the lithium-ion distribution in electrode materials is uneven, resulting in some lithium ions becoming trapped in electrodes such that they cannot react normally. However, when the battery is in a resting state, the potential inside the battery gradually balances out. At the same time, trapped lithium ions in the electrodes will gradually release, thus temporarily increasing the battery’s capacity. This phenomenon is called the capacity regeneration phenomenon or the capacity self-recovery phenomenon. The existence of the capacity regeneration phenomenon means that the battery’s degradation capacity temporarily recovers after a rest period during testing. This brings some challenges to battery degradation modeling and remaining useful life (RUL) prediction. Therefore, this regeneration phenomenon needs to be considered when evaluating battery performance and predicting life.

Generally speaking, there are many factors affecting battery degradation, and most factors affect each other. This makes the lithium battery degradation process complex and nonlinear. However, batteries have more indicators for evaluating their remaining lifespan. For example, voltage, current, temperature, internal resistance, discharge capacity, and time intervals between equal discharge and charge voltage differences. Single or multiple indicators can be used for remaining life prediction.

#### 2.2.3. Turbine Engines

With the development and popularization of transportation equipment, airplanes have become important transportation tools, and safety is the key to aviation. If accidents occur during flight, they will bring disastrous consequences, not only causing huge property losses but also resulting in a large number of casualties. Therefore, the maintenance and repair of aircraft are particularly important. As the core component of aircraft, the engine requires special attention, and the importance of engine safety detection and remaining life prediction is self-evident. With the development of technology, aviation engines have developed from early piston engines to turbofan engines. Better performance means increased structural complexity, and in their own working environment, high pressure, high temperature, high speed, remaining life prediction, and timely maintenance play a key role in the safe flight of aircraft.

As the core of the aviation field, the turbofan engine has a very high level of confidentiality, so it is difficult to obtain real engine degradation monitoring data. However, the NASA Ames Research Center has conducted a large number of simulation experiments on its developed commercial aviation system propulsion simulation platform, CMAPSS, providing a large amount of valuable data for people to study turbofan engines. The basic structure of a simulated turbofan engine is shown in Figure 8.

The components are the following:

Fan: Engine fan module, including fan blades, fan disk, and compressor casing;

Combustor: Combustion chamber, where fuel is added to the cycle to generate heat energy;

N1: Fan;

N2: Core shaft;

LPT: Low-pressure turbine;

LPC: Low-pressure compressor;

HPC: High-pressure compressor;

HPT: High-pressure turbine;

Nozzle.

The operation process of turbofan engines is roughly as follows: air enters the system through the fan and then through the LPC and HPC, and it is compressed into the combustor, where fuel and air are ignited. This makes the high-pressure turbine spray out a more intense airflow and drives the LPT and HPT to rotate, making the nozzle spray out strong airflow backward, providing enough thrust.

It can be seen that the turbofan engine is a large-scale, complex, precision instrument. Its overall performance is closely related to the internal parts of the engine, the external working environment, and the usage mode. Therefore, monitoring data from individual or part sensors cannot effectively reflect the real situation of the engine. It is necessary to monitor the parameters of all sensors to judge the health status of the engine. The main contents of turbofan engine condition monitoring are gas path performance monitoring, oil monitoring, and vibration monitoring. However, different sensor data have different sensitivities to engine degradation. Some sensor parameters do not even change according to degradation and cannot reflect engine degradation. Some sensor parameters fluctuate significantly with degradation and show an obvious relationship with degradation. Due to the different functions and measured parts of sensors, their monitoring data are quite distinct. Some have different dimensions and cannot be directly compared in terms of the degree of sensitivity of each group to degradation. Therefore, all sensor data reflected are definitely high-dimensional, meaning that, in the RUL prediction of engines, excluding or minimizing these adverse factors as much as possible is an important problem for the RUL prediction of engines. Adding RUL labels and performing data normalization are common data processing methods for the RUL prediction of turbofan engines.

## 3. Construction of Health Factors

Because some equipment does not allow for frequent shutdowns during operation, and even when the equipment is allowed to shut down, the initial damage is always in a microscopic range and difficult to measure without professional instruments or complex components, internal faults are difficult to measure without destruction; therefore, it is generally impossible to directly observe the degree of damage to equipment. In addition, with the continuous advancement of sensor technology, modern equipment usually uses multi-modal sensor fusion, and the effective fusion of data information from multiple sensors has important significance for better characterizing the health status of the equipment. Therefore, the construction of a health index (HI) plays an important role in PHM. An effective HI not only helps to promote data visualization and continuously describe the health status of the monitored system but also provides prior information for the next RUL prediction, thereby improving prediction accuracy.

According to different HI construction strategies, it can be divided into direct HI and indirect HI. This section discusses these two methods and introduces the relevant literature on them.

### 3.1. Direct HI

Direct HI is also known as physical HI (PHI), which is based on original monitoring data, under the guidance of domain experts or empirical knowledge, and is directly constructed through simple statistical analysis or feature extraction. It has a certain physical meaning: health value. In the research on equipment RUL prediction, the root mean square is the most widely used PHI. Refs. [41,42] both used the effective value as the PHI to estimate the RUL of bearings. In [18,43,44], kurtosis was used as a monitoring indicator based on equipment vibration signals. The effective value was used as the PHI to predict the RUL of bearings. The kurtosis value extracted from the band-pass filter has also been applied to the RUL prediction of bearings [45]. Some researchers have constructed a PHI through statistical characteristics in the time domain. Singleton et al. [46] used the variance of the time domain vibration signal as the PHI. Elforjani et al. [40] suggested using the signal strength estimation value as the PHI. Medjaher et al. [47] calculated the correlation coefficient between two groups of vibration signals captured in different time periods as the PHI. Li et al. [48] calculated the mathematical morphology mode spectrum of vibration signals as the PHI for monitoring the health state of bearings. Some researchers have extracted new PHIs from frequency domain signals. Deutsch et al. [49] used the fast Fourier transform (FFT) of the vibration signal to construct the PHI to estimate the RUL of bearings. Loutas et al. [50] used spectral flatness as a high-frequency index of bearings. Gebraeel et al. [51] extracted fault frequency and its harmonic amplitude as characteristic parameters and used a BP neural network to predict rolling bearing life. The power density of the gear meshing frequency extracted from the envelope spectrum has also been used as a PHI to predict gear RUL [52]. Hu et al. [53] calculated the average value of frequency amplitude in the spectral band as the PHI of an oil pump.

### 3.2. Indirect HI

An indirect HI is often referred to as a virtual HI (VHI), which is usually obtained using AI methods to fuse or reduce the dimensionality of multiple PHIs or multi-sensor signals. It does not directly represent the actual physical health status but serves as a virtual, digital description used to reflect the degradation trend, health status, or performance changes in equipment or systems. Generally speaking, the construction of a VHI can be divided into non-optimization methods and optimization methods [54]. Baraldi et al. [55] applied the automatic correlation kernel regression (AAKR) model to combine different features into VHIs and reconstructed the indicators of the running components into a weighted sum of their health stage modes. The construction of these VHIs does not have any specific advantageous objectives, so it is regarded as a non-optimization method. Unlike non-optimization methods, the VHI in optimization methods is developed according to certain advantageous attributes required by their respective applications to purposefully improve the performance of their required attributes. Among them, principal component analysis (PCA) is one of the most commonly used dimensionality-reduction techniques and is often applied to the construction process of VHIs, which is an effective method to extract basic degradation characteristics. Widodo et al. [56] used PCA to reduce the dimensionality of the feature set and further calculated the difference between the unknown state and the healthy state as the VHI. Zhao et al. [57] adopted PCA and linear discriminant analysis for two-step supervised dimensionality reduction, and then used a simple multivariate linear regression model to estimate the RUL. In [19], the kernel PCA (KPCA) method was used to obtain the covariates of the Weibull proportional hazard model and to predict the RUL of rolling bearings. Fang et al. [58,59] discovered the cross-correlation information of signal streams using multivariate function PCA (FPCA), using the function principal component scores as the VHI. VHIs can also be constructed using two methods: supervised learning and unsupervised learning [60]. In [61], a VHI construction method was proposed based on a convolutional neural network (CNN), while ref. [62] proposed a health index based on a recurrent neural network (RNN-HI). The above studies that constructed VHIs through supervised learning typically assumed that VHIs and RUL are linearly related and require artificial labeling. However, in the actual test process, a nonlinear relationship exists between the VHI and RUL of the equipment, and the method based on unsupervised learning can eliminate the influence of artificial setting labels, which is more conducive to exploring the degradation law of the equipment. Therefore, the construction method based on unsupervised learning is attracting increasing attention from scholars. In [63], a deep CNN autoencoder was used to extract the VHI from original sensor data in an unsupervised way. Peng et al. [64] proposed an unsupervised HI construction method based on a deep belief network. Qiu et al. [65] first proposed a self-organizing map (SOM) as an unsupervised learning method for constructing HIs, and then this technology was widely used in VHI construction. In [66], wavelet packet–empirical mode decomposition (WP-EMD) was used to extract features and SOM to construct a VHI, thus achieving performance degradation evaluation and RUL estimation. In [67], the empirical mode decomposition-self-organizing map (EMD-SOM) method was adopted to analyze vibration signals and construct a VHI. The related literature also includes [68,69]. Although VHI technology has been widely applied in the literature, its application in fault diagnosis and health management (PHM) is still in its infancy and needs further research.

## 4. RUL Prediction Methods Based on AI

In recent years, due to the continuous progress of computer hardware and machine learning algorithms, artificial intelligence research has made great progress. Especially with the rapid development of the Internet of Things, a large number of different types of sensors can easily collect a large amount of historical and real-time data from machine equipment. Artificial intelligence can learn from big data, thereby automatically diagnosing the health status of machine equipment and predicting its future health development trend. Under these circumstances, traditional technology-based diagnosis systems are being replaced by diagnosis systems based on artificial intelligence, thereby improving the efficiency of equipment health management. This section introduces several artificial techniques commonly used in the field of equipment RUL prediction, as shown in Table 3, including artificial neural networks, neuro-fuzzy systems, support vector machines, Gaussian process regression, and fusion methods. It also introduces three common evaluation indicators. Readers can choose the appropriate AI method according to their own research needs.

### 4.1. AI Methods

#### 4.1.1. Artificial Neural Networks

Artificial neural networks (ANNs), also known as neural networks, are complex network structures formed by a large number of connected processing units (neurons), which is a kind of abstraction, simplification, and simulation of the brain’s organizational structure and operational mechanism. An artificial neural network (ANN) simulates the activity of neurons using mathematical models; it is an information processing system established based on imitating the structure and function of the brain’s neural network [70]. It can be divided into multiple layers and single layers, with each layer containing a number of neurons. Different neurons are connected with corresponding weights to form linear or nonlinear classifications. Among them, neurons are the basic components of neural networks, also known as “activation units,” and each neuron can be an independent learning model. Artificial neural networks can model complex nonlinear relationships. In the training process, the connections between units are optimized until the prediction error is minimized and the network reaches the specified accuracy level. Once the network is trained and tested, it can provide new input information to predict the output [71]. Artificial neural networks also have excellent fault tolerance and achieve fast and highly scalable parallel processing [72]. In predicting the remaining life of equipment, artificial neural networks can predict the probability of equipment performance degradation and failure occurrence in a future period based on the equipment’s historical data and current state. In training the network, we can try to combine neural networks with the Weibull distribution to adapt to measurements and avoid fluctuation areas in the time domain [73]. At the same time, to make the prediction more accurate, we can convert the remaining life information into life percentage for prediction; we can also introduce a validation mechanism into the ANN training process to improve the prediction performance of an ANN model [74]. Finally, we smooth the output results; this method can help in equipment maintenance and management, improving equipment reliability and safety [75].

#### 4.1.2. Neuro-Fuzzy Systems

Fuzzy neural systems are a kind of multi-layer forward neural network that combines fuzzy theory with an artificial neural network. When processing information, these systems have a larger processing range and faster information processing speed, so they have a relatively high self-learning and mapping ability. Fuzzy neural systems combine fuzzy theory with artificial neural networks, fully exploiting their respective advantages and making up for each other’s shortcomings [87]. They show excellent ability in dealing with large-scale fuzzy application problems. In the fields of function approximation, control, and classification, neuro-fuzzy systems show great potential [88]. The principle of neuro-fuzzy systems is to use neural networks to represent and optimize fuzzy inference systems so as to realize adaptive learning of fuzzy rules and membership functions. The characteristics of neuro-fuzzy systems are that qualitative (although not precise) knowledge can be modeled, using fuzzy logic to realize the symbolic expression of machine learning. At the same time, neuro-fuzzy systems have the advantages of fast learning speed, strong self-adjustment ability, and low computational complexity [76]. Combined with a heuristic algorithm, we can quickly derive fuzzy rules from a set of training data [78]. Neuro-fuzzy systems show powerful applications in equipment parameter prediction problems [79]. Among them, the Mamdani-type system is more suitable for approximation problems, while the logic-type system may be more suitable for classification problems [80]. Applying neuro-fuzzy systems to RUL is a very novel and promising field, which is worth further research.

#### 4.1.3. Support Vector Machines and Derived Models

1.Support vector machines

In the traditional sense, support vector machines (SVMs) are binary classifiers. Their basic model is defined as a linear classifier with the maximum margin in the feature space, and their learning strategy is to maximize this margin. This can be ultimately transformed into a convex quadratic programming problem—that is, to find a hyperplane that separates the two classes of data as far apart as possible, such that the model can classify new data more accurately. In other words, the classifier is more stable, which is also the biggest difference from the perceptron model. Compared to other machine learning models, SVM models show higher classification accuracy due to their excellent generalization ability and mathematical foundation [16]. In recent years, SVMs have been used not only in an attempt to solve regression prediction problems, but also to solve equipment remaining life prediction problems [81]. For example, in [82], the researchers used SVM models and rolling bearing parameters to conduct equipment remaining life prediction. In [24], the researchers used the data measured by multiple sensors in an aircraft engine for predicting the remaining life of the equipment. It can be seen that using SVMs to conduct remaining equipment life prediction is a future trend. In practical applications, we have many ways to improve the performance of SVM models, such as adding time sequence information to the model [83] or combining the model with the particle swarm optimization method [84]. In short, there are many ways to improve the performance of SVM models in the context of RUL tasks.

2.Support vector regression

Similar to the principle of SVM models, support vector regression (SVR) models are a kind of regression algorithm that uses the idea of support vector machines (SVMs) to fit continuous real data. The goal of SVR models is to find a function that can approach all data points as much as possible within a given error range (called the ε tube). At the same time, SVR models also require the complexity of this function to be as small as possible; that is, the norm of its parameter vector w should be as small as possible. To achieve this goal, SVR models use the Lagrange multiplier method and kernel function technology. The Lagrange multiplier method can transform constraint conditions into penalty terms, thus transforming the optimization problem into an unconstrained dual problem. Kernel function technology can map nonlinear data to a high-dimensional feature space, so that data can be more easily fitted by a linear function in the high-dimensional space. In the RUL field, SVR can deal with small-sample, nonlinear, time series analysis, and other problems. However, there are problems when SVR is applied to kernel parameter selection. Therefore, readers can use the ABC algorithm to optimize SVR kernel parameters [91] or use hybrid chaotic sequences and evolutionary algorithms to determine the appropriate values of its three parameters, which can not only effectively avoid premature convergence (i.e., falling into local optimum), but can also result in SVR models having superior prediction performance [86]. However, SVMs and SVR are point prediction models and do not include probability distribution information, while RVMs make up for this defect.

3.Relevance vector machines

RVMs (relevance vector machines) are a sparse kernel method based on Bayesian learning, which can be used for regression and classification problems. The principle of RVMs is to introduce a Gaussian prior distribution for weight parameters and then estimate hyperparameters and posterior distribution by maximizing the marginal likelihood function. The advantage of RVMs is that they can provide the mean and variance of prediction values, thus reflecting the uncertainty of prediction. Moreover, compared to SVMs, RVMs can produce sparser models and improve the test speed. The disadvantage of RVMs is that their computational complexity is still high, and they are sensitive to kernel function selection. In [87], the researchers combined broad learning systems (BLSs) with relevance vector machines (RVMs) for experimental verification and compared their accuracy with several common machine learning algorithms. The experimental results showed that BLS–RVM has high prediction accuracy, strong long-term prediction, and generalization ability. In [88], the researchers combined deep learning technology with the RVM algorithm to predict the remaining service life of rotating machinery, and the experiment also achieved good results. This provides us with a way of thinking, that is, using RVMs and other algorithms to combine hybrid algorithms to solve practical problems. Similarly, in the investigation process, we found that there is a huge gap in the application of RVM models in RUL, which needs to be filled by researchers.

#### 4.1.4. Gaussian Process Regression

Gaussian process regression (GPR) is a non-parametric, probabilistic modeling method for regression analysis. It is based on the concept of a Gaussian process and obtains a continuous and smooth function in the input space by fitting the data flexibly. In Gaussian process regression, it is assumed that the relationship between the input and output is described by an unknown, random function. This function is regarded as being sampled from a Gaussian process, which is determined by its mean function and covariance function. Gaussian process regression has a wide range of applications in various fields, especially suitable for small-sample, nonlinear, and noisy regression problems. It can handle small sample sizes and adjust the model complexity adaptively to avoid overfitting. At the same time, Gaussian process regression can also perform parameter optimization and model selection to further improve prediction performance. Gaussian process regression models are widely used in machine learning applications because of their flexible representation and inherent prediction uncertainty measurement [89]. In predicting the remaining life of equipment, Gaussian process regression can be combined with health indicators, which can solve the problem of some equipment parameters being unmeasurable [90]. In addition, Gaussian process regression can also be combined with fuzzy evaluation. Specifically, fuzzy evaluation can be used to preprocess the observed data. Then, by combining fuzzy logic with expert knowledge, overfitting can be avoided in cases of limited data. This method can effectively reflect the uncertainty of prediction results [91]. In the era of rapid improvement in computing power, we can also combine GPR with deep learning technology. For example, we can integrate the LSTM model into the GPR model. First, we can decompose the data into some intrinsic mode functions (IMFs) and residuals using the empirical mode decomposition method (EMD). Then, we can apply LSTM to estimate the residuals and use a Gaussian process regression (GPR) sub-model to fit the IMFs with the uncertainty level while capturing the long-term dependence and uncertainty quantification caused by capacity regeneration [92]. This method also has the advantages of fast computing speed and high accuracy.

#### 4.1.5. Hybrid Methods

Hybrid methods are divided into two types. The first type is a physical model plus a data-driven model. This method is based on the data-driven model, introducing constraint conditions or prior knowledge of the physical model, making the data-driven model more consistent with physical laws, thus improving the prediction performance. For example, using a Bayesian network combined with a physical model for predicting the remaining life of equipment [93]. The second type is a pure data-driven model, similar to an ensemble learning model, which combines multiple machine learning algorithms. We mainly discuss the latter here. As we mentioned above, we can combine Gaussian process regression with fuzzy evaluation [91] or combine deep learning technology with RVMs [88]. One advantage of this approach is that it can combine the advantages of multiple models, allowing them to complement each other’s shortcomings, which greatly improves the efficiency of model use.

#### 4.1.6. Discussion on Suitable Attributes for Different Scenarios

##### Small-Scale Datasets

For small-scale datasets, the key is to select algorithms with low data requirements and moderate model complexity to avoid overfitting while maintaining good generalization ability. SVMs, SVR, RVMs, and GPR are ideal choices. These algorithms perform well on small datasets, especially GPR and RVMs, which can effectively learn from limited data. By optimizing decision boundaries and kernel functions, SVMs and SVR also achieve good results on small datasets. In contrast, due to their higher model complexity and data requirements, ANNs and hybrid methods may need more data to avoid overfitting and are, thus, less suitable for small-scale datasets. Additionally, neuro-fuzzy systems, although flexible, may require more data to train the neural network part and generate fuzzy rules, making them less suitable than the aforementioned algorithms.

##### Noisy Data

When dealing with noisy data, the algorithm’s noise sensitivity is a crucial attribute. GPR, with its noise-modeling capabilities, is the most robust to noise and can effectively handle noisy data. SVMs and SVR, through kernel functions and regularization techniques, can also resist noise to some extent, but not as effectively as GPR. ANNs and neuro-fuzzy systems are more sensitive to noise and are prone to performance degradation due to interference from noisy data. RVMs and hybrid methods have noise-handling capabilities that depend on their specific implementations, but overall, they are less stable than GPR and SVMs in noisy data scenarios. Therefore, in noisy data scenarios, GPR is the top choice, followed by SVMs and SVR. These algorithms not only have good robustness to noise but also maintain a good balance in terms of model complexity and prediction speed.

##### Real-Time Applications

Real-time applications have strict requirements for prediction speed and training time. ANNs, SVMs, SVR, and RVMs have fast prediction speeds, which meet the needs of real-time applications. In particular, SVMs and SVR are highly suitable for real-time scenarios due to their efficient prediction mechanisms. GPR has a relatively slower prediction speed and may not be suitable for highly time-sensitive, real-time applications. Neuro-fuzzy systems have a medium prediction speed, depending on their structure and implementation. Hybrid methods have medium prediction speeds, as they may involve integrating prediction results from multiple models, making their real-time performance less stable than that of single algorithms. In terms of training time, SVMs, SVR, and RVMs have relatively shorter training times, allowing for quick model updates and suitability for real-time applications. ANNs have a longer training time and may not be suitable for scenarios requiring frequent model updates. GPR also has a long training time and is not ideal for real-time updates. Due to their need to separately train each sub-model and perform fusion optimization, hybrid methods have a longer overall training time and should be used cautiously in real-time applications. In summary, for real-time applications, SVMs and SVR are the best choices, as they meet the requirements for prediction speed and training time in real-time applications and also perform well in terms of model complexity and generalization ability.

### 4.2. Evaluation Index

Establishing a unified metric is of great significance for comparing RUL prediction methods, and these metrics can be proposed according to the different requirements of researchers and operators. This section only lists three evaluation metrics that are more commonly used in the literature.

1.Mean absolute error

The mean absolute error (MAE) is a measure of model performance that considers the L1 distance between the true value and the predicted value, also known as the L1 norm loss, defined as follows:(4)$$\mathrm{MAE}(y,y^{\prime })=\frac{1}{m}\sum_{i=1}^{n}|y_{i}-y_{i}^{\prime }|$$

The mean absolute error solves the problem of positive and negative cancellation in the residual sum and provides a better measure of the regression model’s performance. However, because it involves absolute values, the resulting functions are non-smooth and cannot be differentiated at some points. In other words, the mean absolute error is not second-order continuously differentiable, and the second-order derivative is always 0.

2.Mean absolute percentage error

The mean absolute percentage error (MAPE) and the mean absolute error have no second-order derivatives. However, unlike the mean absolute error, the mean absolute percentage error not only considers the error between the predicted and true values but also considers the ratio between the error and the true value. The definition of the mean absolute percentage error is as follows:(5)$$MAPE\left(y,y^{\prime }\right)=\frac{1}{m}\sum_{i=1}^{n}\frac{\left|y_{i}-y_{i}^{\prime }\right|}{y_{i}^{\prime }}$$

3.Root mean square error

The root mean square error (RMSE) is a more common evaluation metric. The RMSE can reflect the error size between the predicted and true values. It is expressed in the same unit, which can be easier to explain and understand. The RMSE is more sensitive to larger errors. It provides larger penalties, which can better distinguish model performance in situations where large errors are not expected. The definition of the root mean square error is as follows:(6)$$RMSE\left(y,y^{\prime }\right)=\sqrt{\frac{1}{m}\sum_{i=1}^{n}{\left(y_{i}-{y^{\prime }}_{i}\right)}^{2}}$$

## 5. Case Study

In this subsection, based on the scenario of small sample data, different artificial intelligence methods are used to predict RUL, and then their applicability is compared and analyzed. In this case, three artificial intelligence methods that are more popular, as described in the research comparison in Section 4.1, were used, namely, LSTM in the ANN method, SVR in the SVM method for solving regression problems, and the PF-LSTM fusion algorithm in the fusion method. In this case, the lithium battery research dataset of the University of Maryland and the lithium battery aging dataset of NASA were selected to predict the RUL of lithium batteries.

### 5.1. Dataset Introduction

1.University of Maryland lithium-ion battery research dataset

The University of Maryland’s lithium-ion battery research dataset was collected for battery aging and remaining useful life (RUL) prediction research, specifically for battery performance analysis. The dataset includes experimental data from various batteries, covering charge and discharge conditions, aging processes, fault modes, and more, under different operational conditions. The main features of the dataset include the following:

Data volume and source: This dataset comprises data from various types of lithium-ion batteries (e.g., from different manufacturers and models). It typically includes records of hundreds to thousands of charge–discharge cycles, covering experimental data from multiple batteries.

Key parameters: This dataset records various key parameters related to battery performance, including the following:

Voltage: Voltage data during the charge and discharge processes.

Current: Current data during different charge and discharge stages.

Temperature: Temperature variation inside the battery.

Capacity: Discharge capacity of the battery, used to assess the battery’s health.

Internal resistance: Changes in the battery’s internal resistance, which are closely related to the aging process.

Data recording method: This dataset records the battery performance under different environmental conditions, such as temperature, humidity, and various load conditions. These factors all influence the battery’s performance and rate of aging.

Research purpose: The primary purpose of this dataset is to establish aging models and remaining useful life prediction models for lithium-ion batteries.

2.NASA lithium-ion battery aging dataset

The NASA lithium-ion battery aging dataset focuses more on the performance degradation of batteries under specific cycling conditions, with the goal of developing and validating battery aging prediction models. This dataset contains aging process data of several lithium-ion batteries, aimed at providing scientific insights for battery life management and optimization of battery use.

Data volume and source: This NASA dataset typically includes long-term cycle data from multiple lithium-ion batteries under different load and usage conditions. The battery aging data records the entire process from the new battery status to the final failure. The dataset includes experimental data from multiple batteries, covering thousands of charge–discharge cycles.

Key parameters: This dataset also includes a variety of aging-related data, mainly including the following:

Voltage: Records the voltage variations over time during the charge and discharge processes.

Current: Records the charging and discharging current, reflecting the battery’s operating state.

Capacity: The charging and discharging capacity, which helps identify the battery’s health status and degradation rate.

Internal resistance: The internal resistance of the battery is a key indicator of aging, which typically increases over time.

Temperature: Temperature significantly impacts the performance and lifespan of lithium-ion batteries, so temperature data are frequently recorded.

Experimental conditions: The experiments in this dataset cover battery usage under different loads, temperatures, and charging strategies. By simulating various real-world environments (high-temperature environments, overcharging, over-discharging, etc.), the aging process and performance changes of batteries under different conditions can be analyzed.

### 5.2. Research Introduction

In this case, we selected the degradation datasets of batteries No. 3 (A3), No. 5 (A5), No. 8 (A8), and No. 12 (A12) in a group of lithium battery cycle experiments in the University of Maryland lithium battery dataset to simulate a small sample data scenario. We also selected the degradation datasets of batteries No. 5 (B0005), No. 6 (B0006), No. 7 (B0007), and No. 18 (B0018) in a group of lithium battery cycle experiments in the NASA lithium battery dataset to simulate a small sample data scenario. We then selected the battery capacity of the two datasets as health indicators and extracted the battery capacity from the datasets. The relationship curve between battery charge–discharge cycle times and battery capacity is shown in Figure 9 and Figure 10.

First, in this case, we uniformly set the failure threshold to 70% of the rated capacity and used the following general processes to predict RUL using various methods:Based on the battery capacity curve 1, we chose the LSTM algorithm as the prediction method for lithium battery RUL. According to the theoretical basis and model-building method of the LSTM algorithm, we used the battery capacity data in A12 and A3 as a training set to train the model, and we used the capacitance data in A5 and A8 before 100 cycles as a test set for prediction. The prediction starting point was set at 100 cycles.Based on battery capacity curve 1, we chose the PF-LSTM fusion method as the prediction method for lithium battery RUL. According to the theoretical basis and model-building method of the PF and LSTM algorithms, we used the same prediction process as in 1.Based on battery capacity curve 2, we chose the SVR algorithm as the prediction method for lithium battery RUL. According to the feature extraction and SVR algorithm model-building method mentioned in [1], we respectively took 40% and 50% of the constant pressure rise time series and constant pressure drop time series as the X label. We then took the battery capacity at the corresponding cycle times as the Y label, and we used the X and Y labels as inputs for the SVR model training. We used 60% and 50% of the cycle data as test sets for prediction, with the prediction starting points set at 40% and 50% of the total cycles, respectively.

### 5.3. Experimental Method

In Section 5.1 of the lithium battery case study, we set the failure threshold of the lithium battery to 70% of its rated capacity. Based on this threshold, we designed various methods to predict the remaining useful life (RUL) of the battery. To ensure reproducibility, the following provides a more detailed explanation of the methodology.

First, we selected the long short-term memory (LSTM) algorithm as the primary model for predicting the RUL of lithium batteries, using capacity curves for the prediction. The training data were sourced from the capacity data of the A12 and A3 batteries, and the testing data came from the A5 and A8 batteries, with RUL prediction starting from cycle 100. In the experiment, we divided the dataset into a training set and a test set. The training set was used for model fitting, while the test set was only used for evaluating model performance. During validation, we used the root mean squared error (RMSE) to evaluate the model’s performance across different folds to ensure its stability. To optimize the hyperparameters of the LSTM model (such as learning rate, batch size, and the number of hidden units in the LSTM layer), we employed a grid search method. Additionally, the experiment was repeated five times to ensure result stability, and the standard deviation was calculated to assess the consistency of the predicted results. Finally, RMSE and mean absolute error (MAE) were used to evaluate the model’s prediction accuracy.

Building on this, we further employed the PF-LSTM (particle filter–long short-term memory) model for RUL prediction. PF-LSTM combines the advantages of particle filtering with LSTM, allowing particle filtering to further refine the predictions made by LSTM, thereby improving the accuracy of the RUL prediction. The training data were, again, sourced from the A12 and A3 batteries, and the testing data came from the A5 and A8 batteries, with prediction starting from cycle 100. As with the LSTM model, the dataset was split into a training set and a test set, and five-fold cross-validation was used to evaluate the model across multiple data subsets. The hyperparameters of both the LSTM and PF components (such as particle count, prediction range, and filtering steps) were optimized using grid search to ensure the model’s performance was maximized. The experiment was also repeated five times, and the standard deviation and 95% confidence intervals were calculated to analyze the stability of the predictions. Finally, the performance of the PF-LSTM model was compared to that of the LSTM model to evaluate its advantages in RUL prediction.

For a more comprehensive comparison, we also selected support vector regression (SVR) as a baseline model, predicting based on features extracted from the constant voltage charge and discharge sequences. First, we selected 40% and 50% of the segments from the constant voltage charge and discharge time series as input features, and the corresponding battery capacity was used as the output label. The training set was created from the extracted features and corresponding RUL values, with 60% of the data used for training and the remaining 40% for testing. The test data used 50% and 60% of the cycle data as input, starting predictions from 40% and 50% of the cycles. The dataset splitting method was consistent with the other models. For the SVR model, we also employed five-fold cross-validation to ensure evaluation across multiple data subsets and optimized the SVR model’s parameters (such as kernel function type and regularization parameters) using grid search. Furthermore, the SVR model underwent five repetitions of experiments, calculating the RMSE, MAE, and prediction standard deviation to assess model consistency. Finally, we performed statistical tests to compare the performance of the SVR, LSTM, and PF-LSTM models.

To thoroughly analyze the performance differences among the LSTM, PF-LSTM, and SVR models, we conducted a rigorous statistical comparison, focusing on the differences in the RMSE and MAE. By calculating the 95% confidence intervals for each model, we assessed the reliability of their performance.

In conclusion, through the comparative analysis of the LSTM, PF-LSTM, and SVR methods, we ensured the reliability and stability of the results. Through cross-validation, hyperparameter tuning, and statistical testing, this study provides a reference model evaluation framework for future lithium battery RUL prediction.

### 5.4. Results Comparison

In this case, we uniformly selected the RMSE (root mean square error) and MAE (mean absolute error) in Section 4.2 as accuracy evaluation indicators for the RUL prediction model. We also defined Ea and Er as the absolute error and relative error between the actual RUL and predicted RUL of lithium batteries, respectively, as follows:(7)$$Ea=\left|PRUL-RUL\right|$$(8)$$Er=\left(\frac{Ea}{RUL}\right)\cdot 100\%$$

The prediction results of the LSTM algorithm are shown in Figure 11, the SVR algorithm prediction results are shown in Figure 12 and Figure 13, and the PF-LSTM algorithm prediction results are shown in Figure 14. A comparison of the results of the three methods is shown in Table 4.

The green line in Figure 10 and Figure 13 represents the failure threshold, the red curve represents the predicted battery capacity value, the blue curve represents the actual battery capacity value, and the black vertical line represents the prediction starting point.

PRUL in Figure 11 and Figure 12 represents the predicted remaining useful life value, RUL represents the actual remaining useful life value, the green horizontal line represents the failure threshold, the red curve represents the actual battery capacity value, the blue curve represents the predicted battery capacity value, and the black vertical line represents the prediction starting point.

### 5.5. Summary

It can be seen from Table 4 that, under the scenario of small sample data, based on the LSTM algorithm, the RUL prediction results are not very accurate, while for the SVR algorithm—which is good at dealing with small sample data—the RUL prediction accuracy is greatly improved. However, when setting different prediction starting points, the results show that predictions made from later starting points yield better RUL prediction accuracy than those from earlier starting points, indicating that SVR is more accurate for dealing with later-stage temporal data. Additionally, the RUL prediction results based on the PF-LSTM fusion algorithm demonstrate that combining the PF algorithm’s strength in modeling degradation mechanisms with the ability of LSTM to handle temporal problems can indeed leverage their respective advantages, improving the overall prediction performance. This greatly improves the prediction accuracy and clearly shows that fusion algorithms can effectively enhance their respective advantages and improve the overall prediction performance. This case study used different artificial intelligence algorithms to predict RUL under the same scenario, and the results prove that there are differences in the effect of artificial intelligence methods when facing different types of equipment or different scenarios and carrying out remaining life prediction applications. Only by clearly mastering the characteristics of various artificial intelligence algorithms, as well as the objects and scenarios they are suitable for, can we achieve good application effects and extract the optimal value of advanced algorithms.

## 6. Conclusions and Future Challenges

### 6.1. Conclusions

This study divided the RUL prediction process into three main technical stages—namely, data acquisition, HI construction, and RUL prediction—and reviewed the current state of equipment RUL prediction research. In the data acquisition stage, we discussed data classification and equipment data; in the HI construction stage, we discussed PHI and VHI from two perspectives; in the RUL prediction stage, we identified five common AI techniques. It should be noted that there is a considerable amount of literature on this topic, covering a variety of types. As we could not review all of it, some research may have inevitably been omitted. Despite the significant progress made in the field of equipment RUL prediction, there are still many challenges.

### 6.2. Challenges

With the rapid development of modern technology, mechanical equipment is becoming more diverse and complex, and the equipment development in aerospace, manufacturing, automation, electronic information technology, new energy, and other fields is moving in a multi-functional direction. Most equipment needs more precise and stable operation, making the fault prediction and health management (PHM) of equipment ever more important, with higher demands placed on the accuracy of remaining useful life prediction results.

#### 6.2.1. Domain Adaptation Challenges

In engineering fields such as mechanical manufacturing, aerospace, and power systems, the operating environments and conditions of equipment are highly diverse. Although AI models perform well on general datasets, their application in specific engineering environments still requires fine-tuning for specific equipment and operating conditions. These specific environments often rely on complex engineering parameters and contextual information, and the generalization ability of models is limited in the absence of target data. The fine-tuning process faces challenges such as high computational costs and a lack of labeled data, especially for specific types of equipment. For example, in the aerospace field, the RUL prediction of aircraft engines needs to consider specific flight conditions and maintenance history, which are often difficult to obtain and have high labeling costs. In addition, under open-domain conditions, the generalization ability of models with respect to unseen equipment types or operating environments is insufficient. Even if they perform well on training data, the prediction accuracy in practical applications may be very low. This indicates that there are significant differences in the adaptability of models in different engineering fields, and more efficient data augmentation methods are needed to improve the adaptability of models. Researchers have explored several data augmentation methods to improve the adaptability of models and clarify the relationships between different data augmentation schemes and specific dataset shift types through evaluation sets. In addition, a generalization test has been proposed that can estimate the shift type of the target dataset and predict which data augmentation scheme is most effective for domain adaptation without training the target domain model.

#### 6.2.2. AI Model Interpretability Challenges

In the engineering field, the interpretability of AI models is a key factor in ensuring their widespread acceptance and trust in practical applications. Advanced machine learning models, such as deep neural networks, are inherently complex and nonlinear. Their high-dimensional representations make it difficult to track the relationships between inputs and outputs. There is often a trade-off between model performance and interpretability, with simpler models (such as linear regression or decision trees) being easier to interpret but not achieving the performance levels of complex models. There is currently no global standard for evaluating the interpretability of machine learning models, and different methods can provide different explanations for the same model, making it difficult to compare and validate them without a formal framework. In addition, interpretability is often subjective. What is an easy-to-understand explanation for one engineer may not be sufficient for another. Post-hoc explanations (such as LIME and SHAP) can provide useful insights but should not be regarded as a panacea for ensuring interpretability. These methods may not fully reveal the internal workings of models, especially for complex deep learning models. Most high-precision AI models operate as black-boxes, which limits their applicability in applications that require real-time explanations. Ensuring that explanations are computationally feasible in real-time applications is a major obstacle, especially in the engineering field, where real-time decision-making is often critical.

#### 6.2.3. Uncertainty Quantification Challenges

Uncertainty quantification is key to improving the reliability and credibility of AI models in engineering applications. In fields such as mechanical manufacturing and power systems, dealing with uncertainties in deep learning prediction tasks is an open scientific question. Advances in uncertainty quantification will guide us to adopt more robust prediction architectures, find more effective ways to deal with noisy and incomplete data, and increase confidence in the prediction results obtained from complex engineering systems. There is a need to adopt ensemble methods that follow Bayesian approaches—that is, ensemble techniques that can capture both aleatoric and epistemic uncertainties—in order to address issues such as excessive parameters, optimization biases, and model misspecifications. Before applying uncertainty quantification methods to practical engineering processes, it is necessary to address the increased computational costs and methodological gaps. Uncertainty quantification must be embedded in existing deep learning architectures without increasing computational overheads. In real-time engineering settings, combining uncertainty quantification with interpretable models remains an underexplored area that can enhance the reliability and interpretability of AI models, especially in scenarios requiring real-time decision-making.

#### 6.2.4. Integration of Physics-Based Models Challenges

Integrating physics-based models with AI models is an important way to improve model generalization and prediction accuracy. In the engineering field, such as mechanical manufacturing and aerospace, the full adoption of machine learning and deep learning still faces challenges, especially in terms of model generalization related to changes in equipment types, operating conditions, and maintenance history. In future applications, the integration of machine learning into the engineering field will follow a hybrid modeling approach, combining data-driven techniques with traditional physics-based methods. Physics-driven and reduced-order models will be used to improve the computational efficiency and accuracy of complex engineering system simulations. There is a need to develop standardized benchmarks, theoretical foundations, and scalable algorithms that can be easily integrated into engineering workflows to enhance the reliability and recognition of deep learning models within the engineering community.

In the field of RUL prediction for engineering equipment, domain adaptation, AI model interpretability, uncertainty quantification, and the integration of physics-based models all face their own challenges. These challenges not only affect the performance and reliability of the models but also affect their acceptance and trust in practical engineering applications. Future research needs to make breakthroughs in these areas to improve the generalization ability, interpretability, uncertainty, and integration of physical models, thereby promoting the widespread application of AI technologies in equipment health management and predictive maintenance.

## Figures and Tables

**Figure 1 sensors-25-04481-f001:**
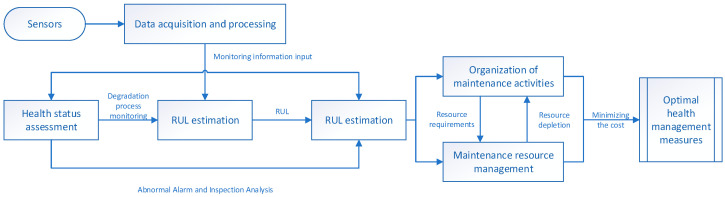
Basic components of PHM.

**Figure 2 sensors-25-04481-f002:**
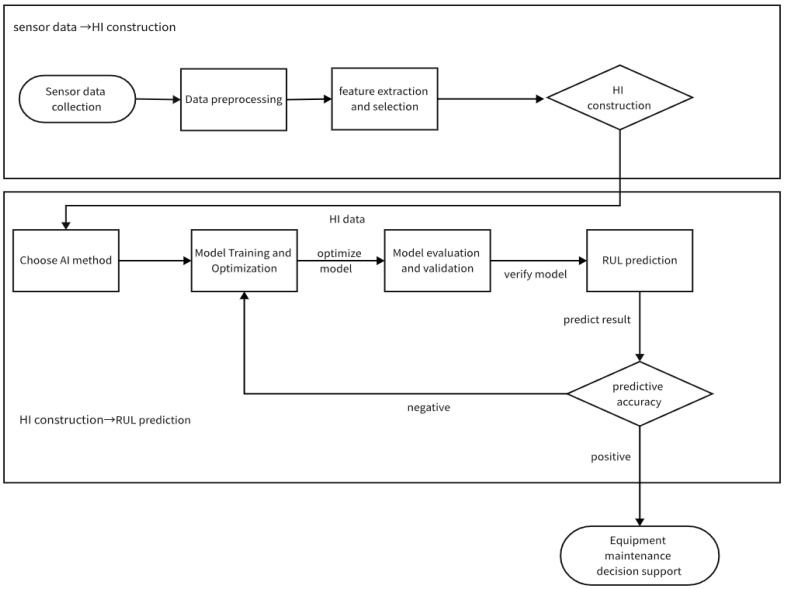
Framework diagram of RUL.

**Figure 3 sensors-25-04481-f003:**
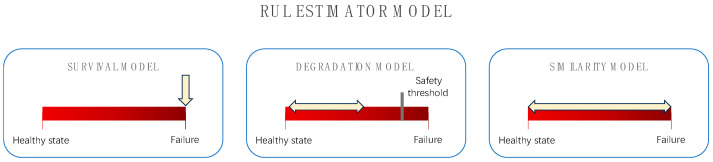
Correspondence between the model and data.

**Figure 4 sensors-25-04481-f004:**
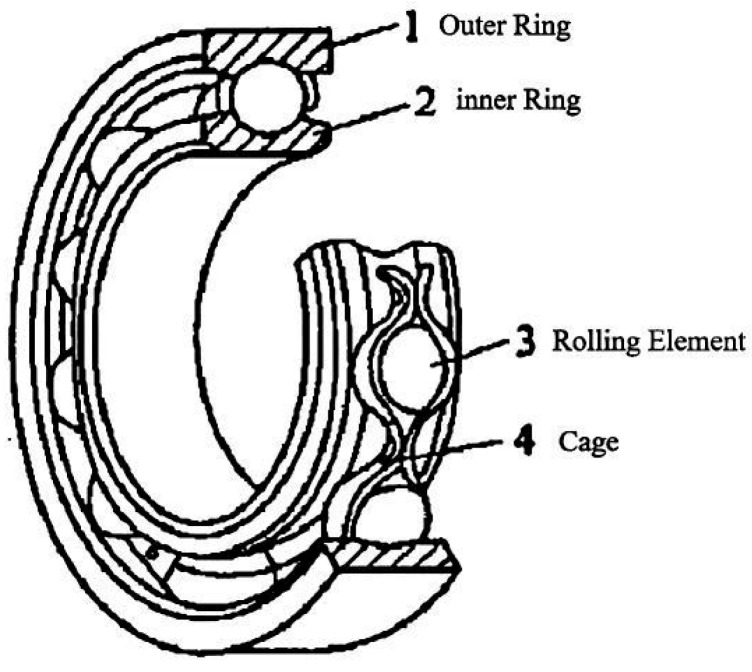
Basic structure of rolling bearings.

**Figure 5 sensors-25-04481-f005:**
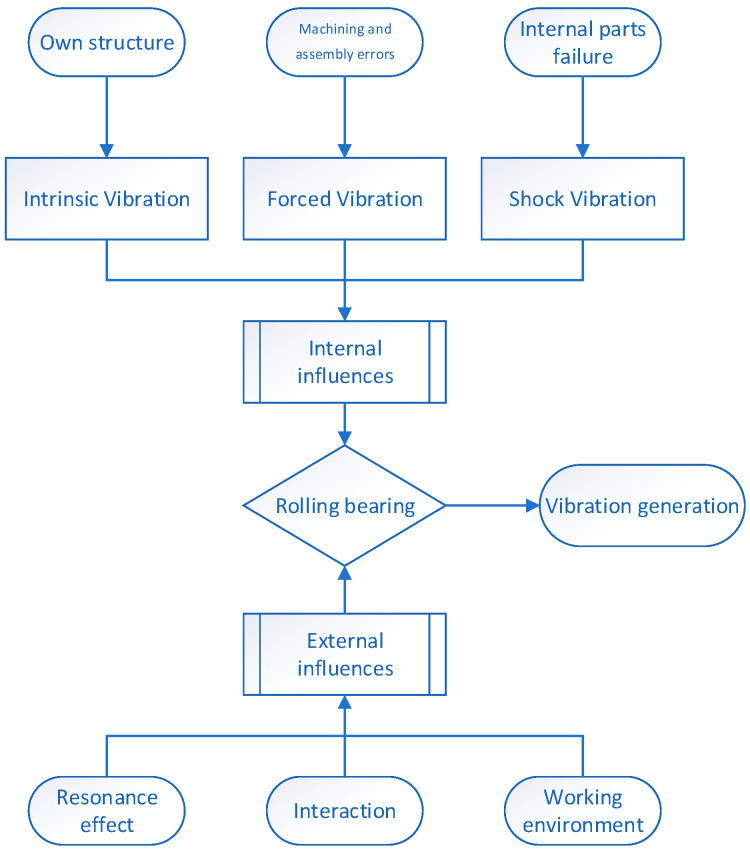
Diagram of vibration-generating factors.

**Figure 6 sensors-25-04481-f006:**
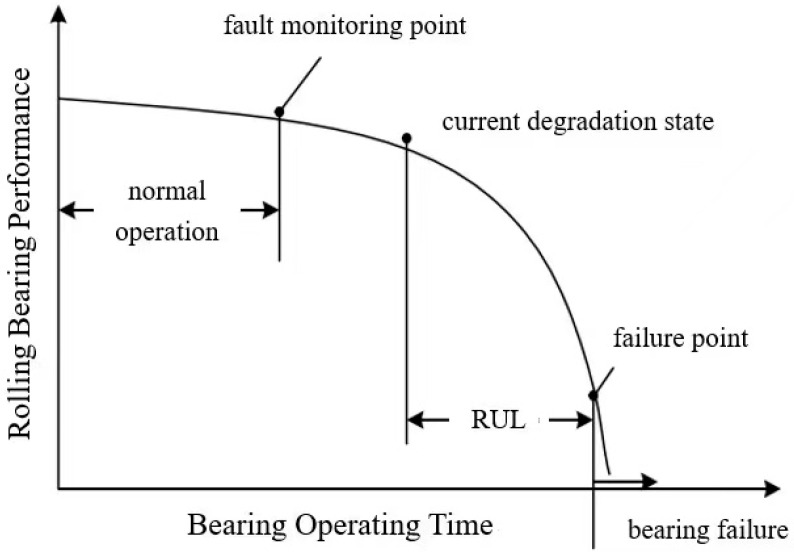
Rolling bearing operating stages.

**Figure 7 sensors-25-04481-f007:**
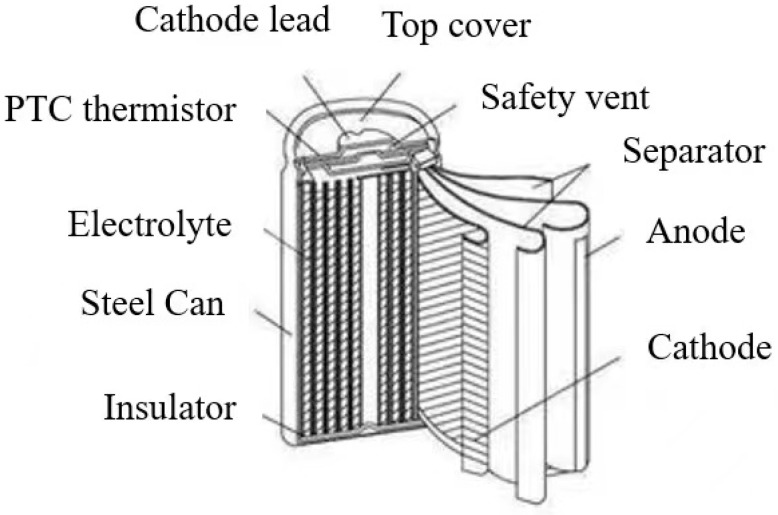
Internal structure of a lithium battery.

**Figure 8 sensors-25-04481-f008:**
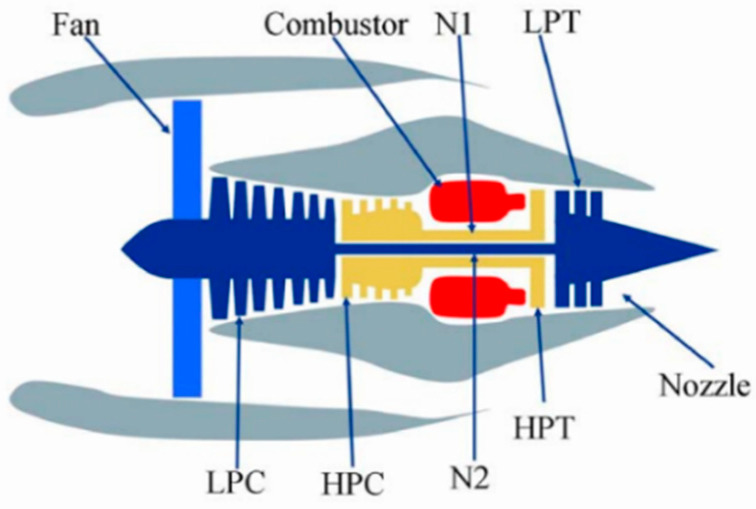
Diagram of the basic mechanism of a turbine engine.

**Figure 9 sensors-25-04481-f009:**
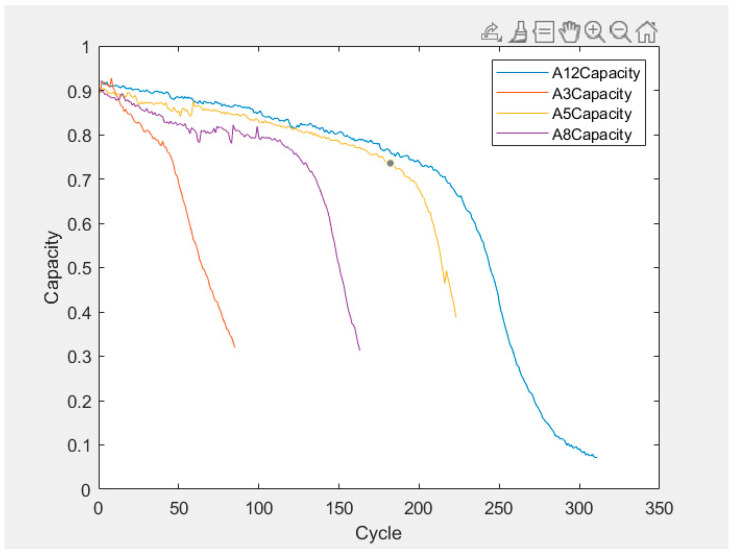
Battery capacitance degradation curve 1.

**Figure 10 sensors-25-04481-f010:**
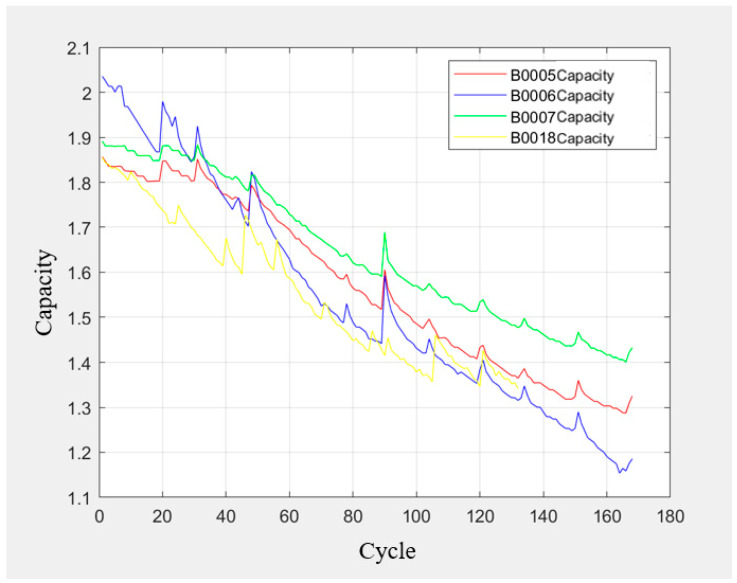
Battery capacitance degradation curve 2.

**Figure 11 sensors-25-04481-f011:**
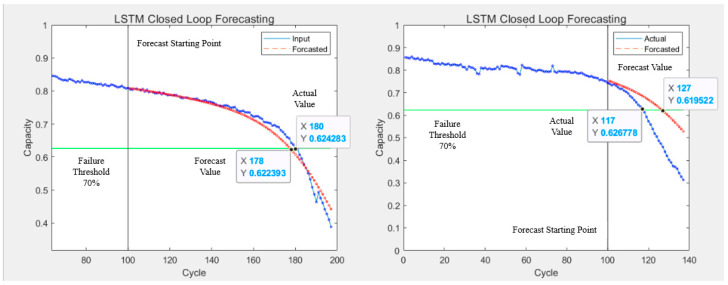
Prediction results using the LSTM algorithm.

**Figure 12 sensors-25-04481-f012:**
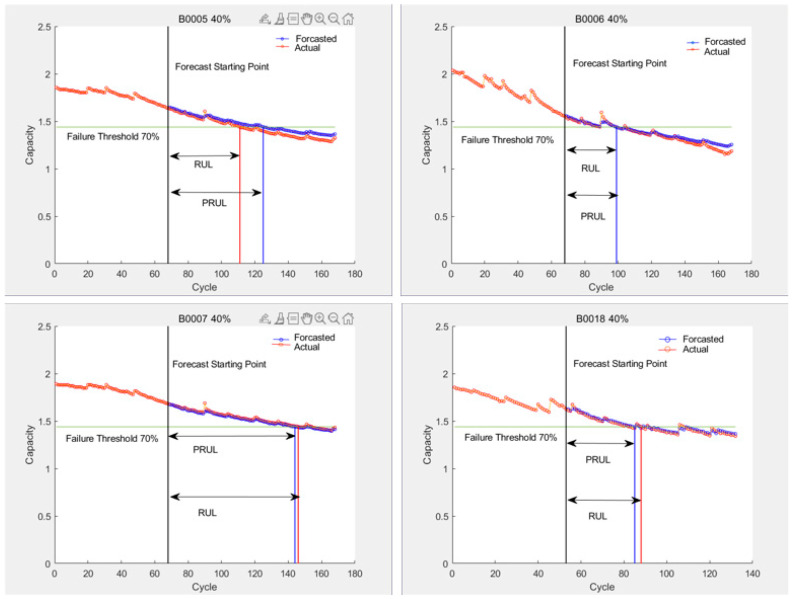
Predicted results using the SVR algorithm (40%).

**Figure 13 sensors-25-04481-f013:**
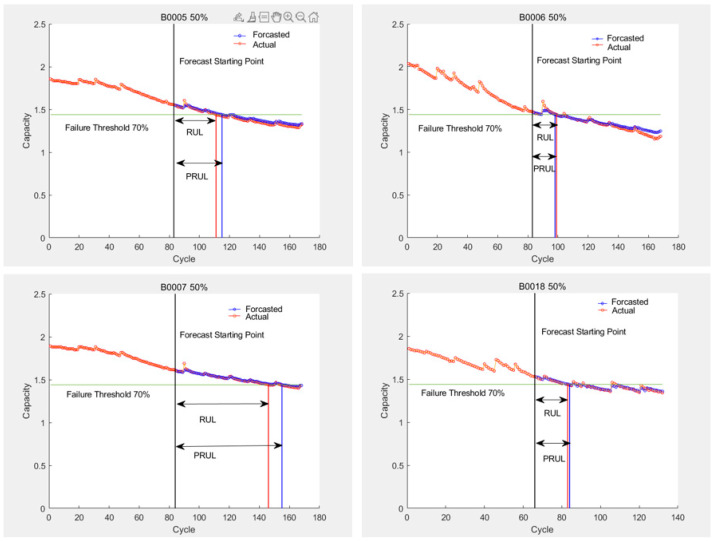
Predicted results using the SVR algorithm (50%).

**Figure 14 sensors-25-04481-f014:**
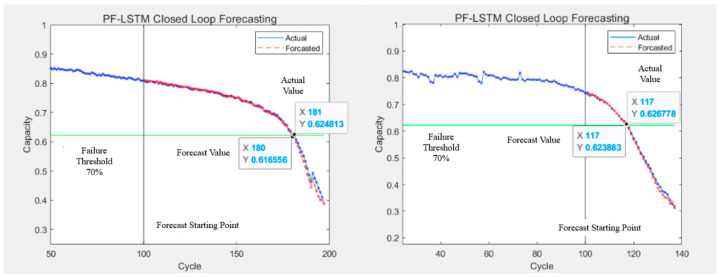
Predicted results using the PF-LSTM algorithm.

**Table 1 sensors-25-04481-t001:** Data classification table.

Data Type	Data Volume	Applicable Models	Refs.
Full degradation data	Data for the entire degradation process from normal to failure	Global Similarity Model Local Similarity Model et al.	[2,3,4,5,6,7,8,9,10,11]
Partial degradation data	Partial data from the normal state to the moment of failure and safety thresholds for failure thresholds	Linear Degeneracy Model Exponential Degeneracy Model et al.	[12] [13,14,15,16,17]
Moment of failure data	Fault data and some covariates associated with RUL	Reliability Survival Model Covariate Survival Model	[18,19,20,21]

**Table 2 sensors-25-04481-t002:** Equipment data table.

Installations	Rolling Bearings	Lithium-Ion Batteries	Turbofan Engines
Application areas	Service industry Aerospace Heavy industry	New energy industry Electronic equipment Aerospace	Aerospace
Common working environment	High temperature and pressure Wet or dry Operating at extreme speeds Overloaded	Long-term continuous operation High charge/discharge frequency Installed in automobiles or airplanes Variable environment	High temperature and pressure Operating at extreme speeds Overloaded
Common input features	Vibration frequency Temperature	Voltage Resistance Charge and discharge time Charge and discharge voltage Capacitance	Diverse and full sensor monitoring data
Data characteristics	Small fluctuations in change during the normal phase and a clear trend during the degradation phase Data information contains a lot of noise	Many considerations Lower data volume	High volume and complexity of data Different levels of correlation between data
Key to forecasting	Denoising Filtering out sensitive features rich in degradation information	Selection of single or multiple judgmental features Sensitivity of model parameters to prevent overfitting and underfitting	Removal of redundant information Filtering features related to RUL Ranking features and constructing RUL labels for different degradation modes
Difficulty in forecasting	Phased degradation Mostly requires full-cycle data	Some data difficult to access Data acquisition is easily interfered with Varying environments and operating conditions Complex modeling makes it difficult to ensure accuracy	Data preprocessing Multi-dimensional data feature integration Feature mapping modeling Information gradient vanishing/explosion
Refs.	Chen et al. [37], Ji et al. [38], Cai et al. [39]	Ge et al. [29], Zhang et al. [30], Lipu et al. [31], Elforjani et al. [40], Sharma et al. [23]	Wang et al. [33], Ellefsen et al. [34], Miao et al. [35], Huang et al. [36], Ordonez et al. [24]

**Table 3 sensors-25-04481-t003:** Summary and comparison of AI methods.

Names	Model Complexity	Scalability	Interpretability	Data Requirements	Features	Refs.
Artificial neural network	Medium–high Increases with more hidden layers and neurons	High Can scale well with large datasets and complex models through distributed training and hardware acceleration	Low Neural networks are often considered “black-box” models and are difficult to interpret	High Requires a large amount of data to train neural networks to avoid overfitting	Ability to model complex nonlinear relationships During training, the connections between the cells are optimized until the prediction error is minimized and the network reaches a specified level of accuracy	[70,71,72,73,74,75]
Neuro-fuzzy system	Medium–high Combines neural network and fuzzy system structures, with model complexity depending on the number of layers in the neural network part and the number of fuzzy rules	Medium Can scale with large datasets but may require careful tuning of the neural network part and fuzzy rules	Medium Fuzzy rules provide some interpretability, but the neural network part is harder to interpret	Medium Needs a certain amount of data to train the neural network part and generate fuzzy rules	Fast learning speed Strong self-adjustment ability Low computational complexity	[76,77,78,79,80]
Support vector machine	Medium Depends on the number of support vectors and feature dimensions	Medium Can handle large datasets but may require more computational resources and optimization techniques	Medium The model can be interpreted to some extent through support vectors and decision boundaries	Low–medium Performs well on small-scale datasets but can also benefit from large-scale datasets	Excellent generalization ability and mathematical foundation	[16,24,81,82,83,84]
Support vector regression	Medium–high Similar to SVM, but regression tasks may require more complex kernel functions and parameter tuning	Medium Similar to SVM, with potential scalability challenges for very large datasets	Medium Similar to SVM, the regression function and related parameters can be used to explain the model	Low–medium Similar to SVM, with good adaptability to small-scale datasets	The ability to transform optimization problems into unconstrained dual problems The ability to map nonlinear data to high-dimensional feature spaces	[85,86]
Relevance vector machine	Medium–high Similar to SVM, but may involve more complex optimization processes to determine relevance vectors	Medium Can scale to large datasets but may require significant computational resources	Medium The model can be interpreted to some extent through support vectors and decision boundaries	Low Relevance vector machine works well on small-scale datasets	Can provide the mean and variance of the predicted values Can produce sparser models than SVM	[87,88]
Gaussian regression process	High Bayesian non-parametric model based on kernel functions, with a significant increase in computational complexity as the data volume increases	Low Computationally intensive and may struggle with very large datasets due to its Bayesian nature and need for matrix inversion	Medium The mean function and covariance function of the Gaussian process provide a certain level of interpretability	Low Gaussian process regression performs well on small-scale datasets, but the computational cost is high for large-scale datasets	Ability to handle problems with a small sample size Ability to adapt the complexity of the model to avoid overfitting	[89,90,91,92]
Hybrid method	High Combines multiple algorithms, with model complexity depending on the number and types of algorithms integrated	Medium–high Depends on the scalability of the individual algorithms integrated and the overall system design	Low–medium The interpretability of hybrid methods depends on the algorithms integrated, and the overall model may be more difficult to interpret	Medium–high Depends on the algorithms integrated, as more data may be required to train each sub-model	Combine the benefits of multiple models Can complement each other’s shortcomings among models	[88,91,93,94]

**Table 4 sensors-25-04481-t004:** Comparison of the prediction results of various algorithms.

Algorithm	RUL	PRUL	MAE	RMSE	Ea	Er
LSTM (A5)	80	78	0.0120	0.0168	2	2.50%
LSTM (A8)	17	27	0.1070	0.1229	10	58.90%
SVR (B0005 40%)	111	125	0.0383	0.0417	14	12.61%
SVR (B0006 40%)	99	99	0.0228	0.0325	0	0%
SVR (B0007 40%)	146	144	0.0101	0.0138	2	1.37%
SVR (B0018 40%)	88	85	0.0146	0.0162	3	3.41%
SVR (B0005 50%)	111	115	0.0184	0.0212	4	3.60%
SVR (B0006 50%)	99	98	0.0212	0.0314	1	1.01%
SVR (B0007 50%)	146	155	0.0067	0.0114	9	6.16%
SVR (B0018 50%)	83	84	0.1031	0.0131	1	1.20%
PF-LSTM (A5)	81	80	0.0070	0.0083	1	1.23%
PF-LSTM (A8)	17	17	0.0064	0.0078	0	0%

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
