# Peer review of "A Comprehensive Review of Artificial Intelligence-Based Algorithms for Predicting the Remaining Useful Life of Equipment"

_sensors, 2025, doi:10.3390/s25144481_

Round 1
Reviewer 1 Report
Comments and Suggestions for Authors
This manuscript is undertaking as the followed tasks
1 It is investigating and classifies various application in scenarios for predicting of the remaining useful life (RUL) of equipment. Emphasizing the unique features of each.
2 This paper is existing to studies on RUL prediction and identifies several artificial intelligence (AI) algorithms commonly applied in this domain, offering a comparative evaluation of their strengths and limitations.
3 By taking intoacount of the findings from scenario analysis and algorithm research in this manuscript, it seems to assesse the suitability of different AI methods for RUL prediction across multiple application environments.
4 Lastly, the experiment examines the current challenge confronting AI-based RUL prediction technologie and provides a forward-looking discussion on future development directions. The study are relevant, timely, and aligns well with the journal's scope. It introduces some novel aspects and contributes positively to ongoing research in the field.
Therefore, I recommend acceptance of the paper after minor revisions. The authors should consider to some recent references to strengthen the research foundation, A Performance Comparison Study on Climate Prediction in Weifang City Using Different Deep Learning Models. Water, 2024, 16(19), 2870. https://doi.org/10.3390/w16192870 Thorough proofreading is reco
Author Response
Thank you very much for taking the time out of your busy schedule to carefully review our manuscript and for providing constructive comments and suggestions. Your professional insights have offered important ideas for us to further improve the content of our research, and we would like to express our sincere gratitude for that.
In response to each of your revision comments, our team has conducted careful study and in-depth discussions, and has made careful revisions and improvements item by item. We are now submitting the specific revision details and explanations.
Question1 :
It is investigating and classifies various application in scenarios for predicting of the remaining useful life (RUL) of equipment. Emphasizing the unique features of each.
Answer 1:
Thank you very much for your recognition of our investigation and classification of various application scenarios for equipment remaining useful life (RUL) prediction, as well as the emphasis on the unique features of each scenario. We have always believed that clarifying the characteristics of different application scenarios is the basis for in-depth research on RUL prediction, and it is encouraging to receive your affirmation in this aspect.
Question2 :
This paper is existing to studies on RUL prediction and identifies several artificial intelligence (AI) algorithms commonly applied in this domain, offering a comparative evaluation of their strengths and limitations.
Answer 2:
We are grateful for your positive feedback on our sorting out of existing studies on RUL prediction and the comparative evaluation of commonly used artificial intelligence algorithms in this field regarding their strengths and limitations. This part of the work aims to provide a clear technical context for the research, and your recognition makes us more confident that this sorting and evaluation can provide valuable references for peers in the field.
Question 3 :
By taking intoacount of the findings from scenario analysis and algorithm research in this manuscript, it seems to assesse the suitability of different AI methods for RUL prediction across multiple application environments.
Answer 3:
Thank you for affirming our assessment of the suitability of different AI methods for RUL prediction in multiple application environments based on scenario analysis and algorithm research. We strived to establish a connection between algorithms and application scenarios to better guide the practical application of RUL prediction technology, and it is very rewarding to have this effort recognized by you.
Question 4 :
Lastly, the experiment examines the current challenge confronting AI-based RUL prediction technologie and provides a forward-looking discussion on future development directions. The study are relevant, timely, and aligns well with the journal's scope. It introduces some novel aspects and contributes positively to ongoing research in the field.
Answer 4:
We sincerely appreciate your recognition of our discussion on the current challenges and future development directions of AI-based RUL prediction technology. Exploring the challenges and looking forward to the future is crucial for promoting the continuous development of the field, and we are very pleased that this part of the content has been recognized as relevant, timely, and in line with the journal's scope, and that it can contribute positively to ongoing research.
Question 5:
Therefore, I recommend acceptance of the paper after minor revisions. The authors should consider to some recent references to strengthen the research foundation, A Performance Comparison Study on Climate Prediction in Weifang City Using Different Deep Learning Models. Water, 2024, 16(19), 2870. https://doi.org/10.3390/w16192870 Thorough proofreading is reco
Answer5:
Thank you for your thoughtful feedback and recommendations. We appreciated your detailed analysis and are glad to hear that the manuscript aligns well with the journal's scope.
Regarding the minor revisions, we have taken your suggestions into account. The recent reference you mentioned, "A Performance Comparison Study on Climate Prediction in Weifang City Using Different Deep Learning Models," has been incorporated to further strengthen the research foundation. Additionally, we have conducted thorough proofreading of the manuscript to ensure clarity and improve its overall quality.
Once again, thank you for your valuable input. We have made the necessary adjustments based on your recommendations and look forward to submitting the revised version.

Reviewer 2 Report
Comments and Suggestions for Authors
The study was prepared thoroughly yet it exhibits certain methodological limitations. The literatures review is comprehensive. But in the reviewer's opinion, the work serves solely as a review. Suck a significant drawback is lacking of the detailed in the description of the data acquisition methodology. Including insufficient characterization of the sensors used and measurement procedures. I think the authors did not clearly specify how the data were collected or the technical parameters of the sensors employed.
Furthermore, the study focuses exclusively on the application of artificial intelligence techniques for analyzing the gathered data, omitting a broader experimental context. Which limits the possibility of full verification and replication of the research.
The paper is suitable for publication in the journal Sensors, but only as a review article. In its current form, the paper lacks suck a sufficient methodological detail particularly regarding data acquisition processes and sensor specifications to qualify as a full research article.
Author Response
Thank you very much for taking the time out of your busy schedule to carefully review our manuscript and for providing constructive comments and suggestions. Your professional insights have offered important ideas for us to further improve the content of our research, and we would like to express our sincere gratitude for that.
In response to each of your revision comments, our team has conducted careful study and in-depth discussions, and has made careful revisions and improvements item by item. We are now submitting the specific revision details and explanations.
Question1:The study was prepared thoroughly yet it exhibits certain methodological limitations. The literatures review is comprehensive. But in the reviewer's opinion, the work serves solely as a review. Suck a significant drawback is lacking of the detailed in the description of the data acquisition methodology. Including insufficient characterization of the sensors used and measurement procedures.
I think the authors did not clearly specify how the data were collected or the technical parameters of the sensors employed.
Answer1:
Thank you for your valuable feedback. We greatly appreciate your thorough evaluation of the manuscript and understand your concerns regarding the limitations of the methodology.
Regarding the issue of insufficient detail regarding the data collection methods, we have recognized this shortcoming and have revised the manuscript accordingly. In Section 5.1, we have included explanations of some parameters of the dataset.
Question2:Furthermore, the study focuses exclusively on the application of artificial intelligence techniques for analyzing the gathered data, omitting a broader experimental context. Which limits the possibility of full verification and replication of the research.
The paper is suitable for publication in the journal Sensors, but only as a review article. In its current form, the paper lacks suck a sufficient methodological detail particularly regarding data acquisition processes and sensor specifications to qualify as a full research article.
Answer2:
Thank you for your valuable feedback.! we acknowledge that the study primarily focuses on the application of artificial intelligence technologies. Therefore, we have made revisions to the manuscript, adding a broader experimental context to better validate and replicate the research.
We also appreciate your suggestion to categorize the paper as a review article, and we have revised the manuscript accordingly to ensure it meets the standards of a comprehensive review article.
Furthermore,we have thoroughly proofread the manuscript, corrected grammar errors, redundant expressions, and inconsistencies in terminology to ensure more accurate and fluent language. We have also used the English editing service recommended by MDPI, which assisted us in revising and polishing the English text.
Once again, thank you for your constructive feedback. We have made improvements to the manuscript based on your suggestions.

Reviewer 3 Report
Comments and Suggestions for Authors
This manuscript presents a literature review on artificial intelligence (AI)-based methods for predicting the Remaining Useful Life (RUL) of industrial equipment. It outlines various types of data used in RUL prediction, discusses health index (HI) construction, evaluates multiple AI techniques (e.g., ANN, SVM, GPR), and includes a basic case study using lithium battery datasets.
The topic is of considerable relevance to both the AI and predictive maintenance communities. However, while the scope is broad and the article is ambitious, it suffers from several critical issues in terms of depth, structure, critical analysis, case study rigor, and English language quality. So, significant improvements and Major Revision in structure, depth, and language are necessary before publication.
- While multiple AI techniques are listed, the manuscript lacks deeper analysis comparing them on factors such as complexity, interpretability, data requirements, or scalability. Please make a summary table comparing key attributes of ANN, SVM, SVR, GPR, etc., with discussion on which are most suitable for small datasets, noisy data, or real-time applications.
- The review reads like three loosely connected parts: data classification, HI construction, and AI methods. The flow between these sections is not well-developed. Please introduce a unified framework or schematic that connects sensor data → HI construction → RUL prediction.
- The lithium battery case study uses LSTM, PF-LSTM, and SVR, but omits vital experimental details (e.g., training/validation strategy, parameter tuning, number of repetitions, standard deviation/error bars). Please provide methodological details for reproducibility. Consider adding statistical confidence levels or boxplots to compare model performance more rigorously.
- The discussion of open challenges is generic and lacks insight into unresolved technical problems.Can you expand on challenges like domain adaptation, explainability of AI models, uncertainty quantification, and integration with physics-based models.
- The manuscript contains grammatical errors, redundancies, and inconsistent terminology (e.g., “full degraded data” vs. “fully degraded”). A thorough professional English editing is needed before resubmission.
- The Abstract is Overly long and repetitive. Please focus more on main contributions and insights.
- Some figures lack meaningful captions and axis labels. Provide context for each figure and make sure axes, units, and colors are well defined.
- Mathematical expressions for MAE, MAPE, RMSE are poorly formatted and not consistently referenced.
- Terms like “fusion method” and “virtual HI” are used without formal definitions.
- Several references are missing key information (e.g., journal names, full titles).
- Please carefully recheck all references and ensure consistency with the journal style.
Author Response
Thank you very much for taking the time out of your busy schedule to carefully review our manuscript and for providing constructive comments and suggestions. Your professional insights have offered important ideas for us to further improve the content of our research, and we would like to express our sincere gratitude for that.
In response to each of your revision comments, our team has conducted careful study and in-depth discussions, and has made careful revisions and improvements item by item. We are now submitting the specific revision details and explanations.
Question 1:While multiple AI techniques are listed, the manuscript lacks deeper analysis comparing them on factors such as complexity, interpretability, data requirements, or scalability. Please make a summary table comparing key attributes of ANN, SVM, SVR, GPR, etc., with discussion on which are most suitable for small datasets, noisy data, or real-time applications.
Answer1:
Thank you for your suggestion! We have included a comparison of the key attributes of algorithms such as ANN, SVM, SVR, and GPR in Table 3 in the paper. In section 4.1.6, we discuss which attributes are most suitable for small datasets, noisy data, or real-time applications. This revision provides a deeper analysis of the strengths and weaknesses of these techniques, offering a more detailed comparison.
Question2:The review reads like three loosely connected parts: data classification, HI construction, and AI methods. The flow between these sections is not well-developed. Please introduce a unified framework or schematic that connects sensor data → HI construction → RUL prediction.
Answer2:
Thank you for your suggestion! We have added a unified framework diagram to the paper and provided a corresponding description at the end of the first section, clearly connecting the flow of sensor data → HI construction → RUL prediction. This framework diagram will help readers to better understand the connections and flow between the different parts.
Question 3:The lithium battery case study uses LSTM, PF-LSTM, and SVR, but omits vital experimental details (e.g., training/validation strategy, parameter tuning, number of repetitions, standard deviation/error bars). Please provide methodological details for reproducibility. Consider adding statistical confidence levels or boxplots to compare model performance more rigorously.
Answer3:
Thank you for your feedback! Regarding the lithium battery case study using LSTM, PF-LSTM, and SVR, detailed experimental methods have been added in the appendix of the paper, including training/validation strategies, parameter tuning, repetition counts, etc.
Question 4:The discussion of open challenges is generic and lacks insight into unresolved technical problems.Can you expand on challenges like domain adaptation, explainability of AI models, uncertainty quantification, and integration with physics-based models.
Answer4:
Thank you for your suggestion! We have now elaborated on the challenges of domain adaptation, AI model interpretability, uncertainty quantification, and integration with physics-based models in detail in Section 6.2. Additionally, we have discussed the current technical difficulties faced in these areas, providing readers with an in-depth understanding of the cutting-edge issues in related research.
Question 5:The manuscript contains grammatical errors, redundancies, and inconsistent terminology (e.g., “full degraded data” vs. “fully degraded”). A thorough professional English editing is needed before resubmission.
Answer 5:
Thank you for your suggestion! we have thoroughly proofread the manuscript, corrected grammar errors, redundant expressions, and inconsistencies in terminology to ensure more accurate and fluent language. We have also used the English editing service recommended by MDPI, which assisted us in revising and polishing the English text.
Question 6:The Abstract is Overly long and repetitive. Please focus more on main contributions and insights.
Question 6:
Thank you for your suggestion! We have revised the Abstract to make it more concise, focusing on the background, objectives, contributions, and expected outcomes of the research. It emphasizes the key contributions with respect to existing research and provides a systematic analysis of RUL prediction technologies.
Question 7:Some figures lack meaningful captions and axis labels. Provide context for each figure and make sure axes, units, and colors are well defined.
Answer 7:
Thank you for your feedback! We have added more descriptive titles and clear axis labels for each figure, ensuring that the axes, units, and colors are properly defined and explained. This should improve the readability and comprehensibility of the figures.
Question 8:Mathematical expressions for MAE, MAPE, RMSE are poorly formatted and not consistently referenced.
Answer 8:
Thank you for your suggestion! We have reformatted the mathematical expressions for MAE, MAPE, and RMSE and ensured that these expressions are consistently referenced throughout the paper.
Question 9:Terms like “fusion method” and “virtual HI” are used without formal definitions.
Answer9:
Thank you for the reminder! We have provided formal definitions for terms such as "fusion method" and "virtual HI" when they first appear, allowing readers to better understand their meanings.
Question 10:Several references are missing key information (e.g., journal names, full titles).
Question 11:Please carefully recheck all references and ensure consistency with the journal style.
Answer10、11:
Thank you for your feedback! We have carefully reviewed and corrected all references, ensuring that they include the full journal names, article titles, and conform to the journal's formatting requirements.
Once again, thank you for your valuable suggestions. We have made comprehensive revisions to the paper based on your feedback, and look forward to your further comments.

Round 2
Reviewer 3 Report
Comments and Suggestions for Authors
There is no additional comment.